# Screen-Printing of Functionalized MWCNT-PEDOT:PSS Based Solutions on Bendable Substrate for Ammonia Gas Sensing

**DOI:** 10.3390/mi13030462

**Published:** 2022-03-18

**Authors:** Direk Boonthum, Chutima Oopathump, Supasil Fuengfung, Patipak Phunudom, Ananya Thaibunnak, Nachapan Juntong, Suvanna Rungruang, Udomdej Pakdee

**Affiliations:** 1Division of Physics, Faculty of Science and Technology, Rajamangala University of Technology Krungthep, 2 Nanglinchi Road, Thungmahamek, Sathorn, Bangkok 10120, Thailand; direk.b@mail.rmutk.ac.th (D.B.); chutima.o@mail.rmutk.ac.th (C.O.); supasil.f@mail.rmutk.ac.th (S.F.); 2Division of Energy Technology for Environment, Faculty of Science and Technology, Rajamangala University of Technology Krungthep, 2 Nanglinchi Road, Thungmahamek, Sathorn, Bangkok 10120, Thailand; 3Division of Printing Technology, Faculty of Science and Technology, Rajamangala University of Technology Krungthep, 2 Nanglinchi Road, Thungmahamek, Sathorn, Bangkok 10120, Thailand; patipak.p@mail.rmutk.ac.th (P.P.); ananya.t@mail.rmutk.ac.th (A.T.); nachapan.j@mail.rmutk.ac.th (N.J.); savanna.r@mail.rmutk.ac.th (S.R.)

**Keywords:** screen-printing, multi-walled carbon nanotubes, PEDOT:PSS, gas sensor

## Abstract

Multi-walled carbon nanotubes (MWCNTs) were grown on a stainless-steel foil by thermal chemical vapor deposition (CVD) process. The MWCNTs were functionalized with carboxylic groups (COOH) on their surfaces by using oxidation and acid (3:1 H_2_SO_4_/HNO_3_) treatments for improving the solubility property of them in the solvent. The functionalized MWCNTs (*f*-MWCNTs) were conducted to prepare the solution by continuous stir in poly(3,4-ethylenedioxythiophene):poly(styrenesulfonate) (PEDOT:PSS), dimethyl sulfoxide (DMSO), ethylene glycol (EG) and Triton X-100. The solution was deposited onto a bendable substrate such as polyethylene terephthalate (PET) with a fabricated silver interdigitated electrode for application in a room-temperature gas sensor. A homemade-doctor blade coater, an UNO R3 Arduino board and a L298N motor driver are presented as a suitable system for screen printing the solution onto the gas-sensing substrates. The different contents of *f*-MWCNTs embedded in PEDOT:PSS were compared in the gas response to ammonia (NH_3_), ethanol (C_2_H_5_OH), benzene (C_6_H_6_), and acetone (C_3_H_6_O) vapors. The results demonstrate that the 3.0% *v*/*v* of *f*-MWCNT solution dissolved in 87.8% *v/v* of PEDOT:PSS, 5.4% *v*/*v* of DMSO, 3.6% *v*/*v* of EG and 0.2% *v*/*v* of Triton X-100 shows the highest response to 80 ppm NH_3_. Finally, the reduction in the NH_3_ response under heavy substrate-bending is also discussed.

## 1. Introduction

Nowadays, electronic equipment bending presents a major shift from rigid devices to flexible and stretchable systems. Because of their low-cost, thin and flexible characteristics, printed electronics provide a novel technology for the replacement of traditional inflexible devices. It has many advantages such as lightweight and easy preparation compared with the conventional vacuum deposition and photolithographic patterning methods. For the gas sensor applications, the printed techniques (direct-writing, inkjet-printing, screen-printing, 3D printing) provide a range of time-saving mechanisms and the full potential of sensing signals for application in the scope of gas sensors [1,2,3,4]. Toxic gases are major problems in human health and the environment. The causes of these problems are gases released from various industries during production processes. Ammonia (NH_3_) is known to provide an effect on the human health as an explosive gas. As colorless gas with a distinct pungent smell, it can even lead to suffocation and death if the level of exposure is high. Therefore, monitoring and timely warning is important for settings in industrial factories and other public places. The development of detection technology is ongoing, with the development of simple approaches including electrochemical and semiconductor devices. However, the life-time of electrochemical sensors is limited by the various electrolytes. The semiconductor sensor involves thin metal oxide films with working temperatures over 150 °C [5,6,7,8]. This is a limitation of operation with flexible plastic substrates. Multi-walled carbon nanotubes (MWCNTs), functionalized multi-walled carbon nanotubes (*f*-MWCNTs) and poly(3,4-ethylenedioxythiophene):poly(styrenesulfonate) (PEDOT:PSS) have long been considered a good selection for room-temperature sensing materials of different gases [9,10,11,12,13,14,15,16]. Moreover, the deposition of *f*-MWCNT-PEDOT:PSS as a sensing layer using the inkjet-printed technique has been reported for the enhancement of gas-sensing properties [17]. However, the problem of clogged nozzles in the printer-head is still a main obstruction for the preparation of sensing layers onto the substrates. Screen-printing techniques have been used to solve the abovementioned problems [3]. It is an effective and simple process used to deposit sensing materials onto the substrates for application in the field of gas sensors. It also has many advantages in controlling the thickness and chemical composition of sensing-materials.

In this study, the *f*-MWCNT-PEDOT:PSS-based solution was prepared as a sensing layer for application in gas sensor. The combination of a homemade-doctor blade coater, an UNO R3 Arduino board and an L298N motor driver was presented as a novel system for screen-printing the solutions onto the gas-sensing substrates. The different contents of *f*-MWCNTs embedded in PEDOT:PSS were compared in the gas response to various gases at room-temperature. The performance of the fabricated gas sensor was further evaluated in sensitivity, selectivity, response time, recovery time and drift parameters. Finally, the reduction in the NH_3_ response for the sensor under heavy substrate-bending was also discussed based on a tensile strain effect.

## 2. Materials and Methods

The MWCNTs were grown on a 1.6 × 3.0 cm^2^ stainless steel (304 SS) foil with a thickness of 50 μm by using a thermal chemical vapor deposition (CVD) process. The MWCNTs were synthesized under atmospheric pressure of acetylene (C_2_H_2_), hydrogen (H_2_) and argon (Ar) gases with a flow rate of 160, 200, and 50 sccm, respectively. The 304 SS foils were heated at a fixed temperature of 700 °C, while all gases were fed into a horizontal chamber. The details of MWCNT growth have been reported in a previous work of our group [18]. After the CVD process, the MWCNT powder on 304 SS foil was scraped from the 304 SS substrate using a plastic rod. The powder was heated up to a temperature of 1000 °C under atmospheric pressure of nitrogen (N_2_) gas for 30 min to remove amorphous regions on the MWCNT surfaces. The powder was then ultrasonically immersed in an 80 mL mixture of sulfuric acid and nitric acid (3:1 H_2_SO_4_/HNO_3_) for 2 h. This functionalization was presented for improving the solubility property of *f*-MWCNTs in solvent [17,19]. After the oxidation and acid treatments, the distilled water was employed to rinse contaminations and some remaining acids on the MWCNT surfaces. Before the preparation of the *f*-MWCNT sensing solution, the powder was dried in an oven at 80 °C for 12 h. The dried *f*-MWCNTs were continuously sonicated in 80 mL of deionized water (DI water) for 45 min. The 0.0, 0.5, 2.0, 3.0 and 5.0% *v*/*v* of *f*-MWCNT solutions were stirred in different concentrations of PEDOT:PSS, dimethyl sulfoxide (DMSO), ethylene glycol (EG) and Triton X-100 for 2 h. The concentration details in terms of the volume percentage of all chemicals for preparing the sensing-solutions are defined by the S0, S1, S2, S3 and S4 samples, as shown in Table 1.

It should be noted that the *f*-MWCNTs and PEDOT:PSS were used as the sensing materials. DMSO, EG and Triton X-100 were used as a solvent, a viscosity modifier and a nonionic surfactant, respectively. The *f*-MWCNTs embedded in PEDOT:PSS, DMSO, EG and Triton X-100 in each condition were screen-printed on polyethylene terephthalate (PET) substrates by using a homemade-doctor blade coater controlled with a UNO R3 Arduino board and an L298N motor driver as shown in Figure 1. A stepper motor in a doctor-blade coater was supplied by a 12 V DC power supply in order to rotate the motor to drive the blade movement along the screen-printing path with a speed of 0.5 cm/s. During the screen printing, a 0.3 N perpendicular force was applied on the substrate modulated by a blade and two micrometers. A UNO R3 Arduino board supplied by a 5 V DC power supply was conducted to compile and control all functions in the system through the input and to enable the channels. To prepare the S0, S1, S2, S3 and S4 gas sensors, the S0, S1, S2, S3 and S4 samples were screen-printed onto PET substrates with fabricated 1.0 × 1.6 cm^2^ silver interdigitated electrodes. The sensors were evaluated on the basis of their sensing performances under gas ambient using a flow-through system as shown in Figure 2. The composition of the system consists of an air pump, an exhaust fan and a rotary pump, two flow meters, a circuit board and two ball valves. The sensor was placed into a four-way cross fitting as a test chamber while a voltage divider circuit was also connected to the test chamber. A rotary pump was used to evacuate and remove the remaining gas out of the chamber before supplying the air and test gas into the chamber for the gas-sensing measurement. A laptop operated with LabVIEW software and an NI USB DAQ 6008 device was used to monitor the gas-sensing signals by measuring the resistance of a gas sensor every second. After the gas-sensing measurement, an exhaust fan was used to drain all gases to the outdoors.

## 3. Results and Discussion

After the CVD process, the MWCNTs were grown on a full area of 304 SS foil. Figure 3 shows a photograph of a 1.6 × 3.0 cm^2^ 304 SS foil before (Figure 3a) and after (Figure 3b) CVD process. Surface morphologies of MWCNTs grown on 304 SS foil were characterized by scanning electron microscope (SEM, Quanta 450 FEI) with a working voltage and current of 30 kV and 10 μA. To observe the density of MWCNTs more easily, a Scotch^®^ tape was conducted to remove the MWCNTs from some area of the foil as shown in Figure 4a. The MWCNTs on the 304 SS foils were ultrasonically sonicated in the DMSO solvent and dropped onto a copper grid. It was then inserted into a sample holder of a high-resolution transmission electron microscope (HRTEM, Hitachi HT 7700). The HRTEM was conducted using an accelerating voltage of 200 kV with a current of 60 μA to examine the size of MWCNT diameter. It can be confirmed that the samples are multiwalled carbon nanotubes with a diameter size of ~35 nm, as shown in Figure 4b. The diameter measurements were calculated using an ImageJ software program in five different areas of the sample. It is seen that the average size of the diameter for the MWCNTs was found to be 35 ± 5 nm. The MWCNTs were functionalized with carboxylic groups (COOH) on their surfaces using oxidation and acid (3:1 H_2_SO_4_/HNO_3_) treatments. Furthermore, 0.5 g of functionalized MWCNTs (*f*-MWCNTs) was continuously sonicated in 80 mL DI water for 45 min. The *f*-MWCNT solutions in each condition were then conducted to prepare the conductive solution by continuous dissolution in PEDOT:PSS, DMSO, EG and Triton X-100. The surface morphologies of *f*-MWCNT-PEDOT:PSS at different sensors of S1, S2, S3 and S4 can be seen in Figure 5. As the content of *f*-MWCNTs in the solution increases, the density of PEDOT:PSS tends to decrease. The formation of functional groups on *f*-MWCNT-PE the DOT:PSS surfaces was characterized using a Fourier transform infrared spectrometer (FTIR, Perkin Elmer Spectrum One) as shown in Figure 6. The weak peak at 1625 cm^−1^ might be assigned to the C=C stretching mode of the graphitic layer for MWCNT. This peak is weak because of the symmetry of the dipole moment on the graphitic layer [13,20]. In the case of the spectrum for *f*-MWCNT-PEDOT:PSS, the peaks contain C-S bond at 705, 858 and 946 cm^−1^ [20]. The peaks at 658, 1095, 1412 and 1713 cm^−1^ indicate the stretching mode of S=O, C-O, C-C and C=O in carboxyl stretching modes, respectively. The dual peaks at about 2900 cm^−1^ for the C-H stretching mode might represent contaminations of hydrocarbon in the spectrometer. The broad peak at around 2900 cm^−1^ is responsible for the O-H groups [21]. This indicates the presence of the formation of carboxylic (COOH) groups on the surface of *f*-MWCNTs embedded in PEDOT:PSS. The *f*-MWCNTPEDOT:PSS solution was deposited onto a PET substrate with a fabricated silver interdigitated electrode with a designed screen-printing system.

Figure 7a shows a photograph of a home-made doctor blade coater and its parts. The screen-printed film of *f*-MWCNT-PEDOT:PSS gas sensor before and after peeling the sticker mask can be seen in Figure 7b,c, respectively. The size of *f*-MWCNT-PEDOT:PSS sensing film is 0.6 × 0.8 cm^2^, as shown in Figure 7c. After the screen-printing process, the sensor was placed into the test chamber of the gas-measurement system. The sensor was investigated for its response to NH_3_ and other target gases at room-temperature. Figure 8 shows the resistance changes of S1, S2, S3 and S4 gas sensors under 80 ppm NH_3_ exposure.

More specifically, the gas sensor without *f*-MWCNT content (S0 gas sensor) was further characterized as a comparison. It is seen that the baseline resistances of all fabricated gas sensors are found to be ~830 Ω. Then, all of the dynamic resistances of sensors increase when the sensors are exposed to 80 ppm NH_3_ vapors. However, the resistances of S0, S1, S2 and S3 sensors recover completely to their baseline after the valve of NH_3_ vapors is closed, as can be seen in Figure 8a. For the high *f*-MWCNT content of the S4 gas sensor, it was interestingly observed in Figure 8b that the resistance cannot return to its baseline line, although the NH_3_ flow stopped. The gas sensors were evaluated in terms of their performances using gas response, selectivity, response time, recovery time and drift value. The gas response was defined by Equation (1).
(1)$$S\left(\%\right)=\frac{R_{gas}-R_{air}}{R_{air}}\times 100,$$
where R_gas_ and R_air_ are the gas-sensor resistances in test-gas and dry-air flows, respectively.

The calculated values of gas response for the S0, S1, S2, S3 and S4 sensors under 80 ppm NH_3_ exposure were 14.5, 32.5, 44.6, 66.3 and 67.5%, respectively. It was found that the gas response of all sensors increased with an increasing *f*-MWCNT contents. However, the gas sensor with a high content of *f*-MWCNTs (S4 sample) demonstrated no noticeable return to the baseline resistance. The drift value of resistance for the S4 sensor was ~350 Ω. It is known that this value is not impressive for sensor preparation. Therefore, the optimum *f*-MWCNT content for the fabrication of an effective *f*-MWCNT-PEDOT:PSS gas sensor exposed to 80 ppm NH_3_ is 3% *v*/*v* of *f*-MWCNT solution (S3 sample). The time of the change for the sensor resistance after a gas-sensing cycle is defined as the response time. For all of the sensors, the response time was found to be of a similar value, at ~3.8 min.

The recovery time of the sensor has been further defined by the time of resistance change and recovery to its baseline. It is seen in Figure 8a that the recovery time of S0, S1, S2 and S3 sensors was a duplicate value of ~4.5 min. For the S4 sample, it cannot be indicated in the recovery time due to the fact that the resistance of the S4 sensor does not perfectly return to its baseline. Normally in physisorption, the gas molecules accumulate on the sensing surface due to weak force, known as Van der Waals forces. The chemisorption involves the strong chemical bonding of the adsorbate with the surface of the adsorbent. Therefore, the chemical adsorption requires activation energy for reversibility in nature. For the gas-sensing layer with a low *f*-MWCNT content (3.0% *v*/*v* solution), the physisorption is stronger than the chemisorption processes in case of NH_3_ sensing by *f*-MWCNT-PEDOT:PSS, while the high *f*-MWCNT content (≥5.0% *v*/*v* solution) of the sensing layer presents very strong chemisorption. Therefore, the content of *f*-MWCNTs embedded in PEDOT:PSS has an important effect on the adsorption of NH_3_ molecules. The effectiveness of the S3 gas sensor in handling repeated inlets of NH_3_ gas is shown in Figure 9. It can be seen that the S3 gas sensor presents a good reproducibility of the sensor for 80 ppm NH_3_ exposure with excellent recovery over four cycles. The earlier publication involving the gas sensor has been focused on the reduction in the recovery time and enhancement of the recovery process. The reduction in the recovery time from 48 h to 20 min has been reported by using the combination of heat and a DC electric field to serve the desorption of NH_3_ molecules from MWCNT surfaces [22]. In this work, the *f*-MWCNT-PEDOT:PSS gas-sensor with the optimum condition (S3 gas sensor) presents good performance in terms of the recovery property in dry-air flows without external excitation. The recovery time of the S3 gas sensor after 80 ppm NH_3_ exposure is less than the time reported for the above work.

For the selectivity property, all of the sensors were compared in gas response under 80 ppm NH_3_, 200 ppm C_2_H_5_OH, 200 ppm C_6_H_6_ and 1000 ppm C_3_H_6_O. It can be observed in Figure 10a that the sensors show the highest response to 80 ppm NH_3_, while the gas responses of all sensors exposed to other gases are lower than 10%. Therefore, the PEDOT:PSS and *f*-MWCNT-PEDOT:PSS gas sensors have a good performance in selectivity to NH_3_. The S3 gas sensor, as the best sample, was warranted a more thorough investigation in relation to its sensitivity property. This property has been evaluated by a slope value of a linear relation between gas concentration and the gas response of sensors under target gas. The calculated values of slopes for S0 and S3 sensors were compared. It is seen in Figure 10b that the sensitivity values of the S0 and S3 sensors are 0.20 and 0.97 ppm^−1^, respectively. Therefore, it can be indicated that the *f*-MWCNT contents embedded in PEDOT:PSS leads to a better sensitivity of the gas sensor.

To study the effect of the bending state on the gas response of sensors, the S3 gas sensor was bent to a curvature radius of 3.0 and 0.9 cm, respectively. Furthermore, the sensor with a flat state was also tested as a comparison. Polylactic acid (PLA) filaments were printed in the cylindrical shape with an outer radius of 3.0 and 0.9 cm by a 3-dimensional (3D) printer. As shown in the diagram in Figure 11a, a PLA cylindrical shape was used as a holder for laying a fabricated sensor on its side surface. The sensor was carefully attached to a PLA holder using a Scotch^®^ tape before it was inserted into a test chamber of the gas-sensing measurement system. It should be noted that only one sensor can be tested at a time. After testing the bending sensors, as shown in Figure 11b, the gas responses of the S3 sensor under flat and 3.0 cm bending-radius states did not present an obvious difference. The calculated values of the gas response for two of the sensors are found to be 67.5 and 65.0%, respectively. For the 0.9 cm bending-radius state of the sensor, it was observed that the calculated value of gas response significantly reduced to 29.5%. The reduction in the NH_3_ response for the sensor under heavy substrate-bending is further discussed based on a tensile strain effect.

To understand the tensile strain effects on the gas response of the *f*-MWCNT-PEDOT:PSS gas sensor, the surface morphology of the S3 screen-printed film after already bending test was further investigated. Figure 12 shows the SEM images of *f*-MWCNTs embedded in PEDOT:PSS sensing films (S3 sample) after bending tests with 3.0 cm and 0.9 cm radii. It was observed in Figure 12a that there were some cracks in the bending-film surface. The sizes of the crack gaps for the films after 3.0 and 0.9 cm bending-radius tests were found to be ~1.1 and ~1.5 μm, respectively. After a test of 3.0 cm bending-radius (Figure 12b), there were *f*-MWCNT alignments in the film, which acted as conductive pathways between the gap. For the much larger crack gap of a 0.9 cm bending-radius film (Figure 12c), a lack of *f*-MWCNT pathways was observed. Delamination, channeling and cracking have been reported as important causes for the failure of breakable films on flexible substrates [23,24,25]. When decreasing the stress on the film surface, the tensile strain increased. Therefore, the crack paths on the film surface were generated. This is the most common observation for the polymer film during the bending process. Figure 13 shows the schematic diagram of pathways in electron transports for a *f*-MWCNT-PEDOT:PSS gas sensor under a flat state (Figure 13a), 3.0 cm (Figure 13b) and 0.9 cm (Figure 13c) bending-radius states. The reduction in the NH_3_ response for the sensor under heavy bending has been also discussed, in that the cracks generate permanent changes in the electrical resistance of PEDOT:PSS sensing films. However, the *f*-MWCNTs embedded in the PEDOT:PSS act as additional pathways in electron transport. Therefore, the changes in electrical resistance of the *f*-MWCNT-PEDOT:PSS sensing film with weak bending have little impact on the electrical property and gas-sensing performance. This may provide a reason as to why the gas response of the *f*-MWCNT-PEDOT:PSS gas sensor under flat and 3.0 cm curvature-radius states did not lead to an obvious difference. However, due to the large crack gap in the heavy bending substrate, the sensor under 0.9 cm bending radius presents a low response to 80 ppm NH_3_. This is due to the lack of *f*-MWCNT connectors between the gap. This results in the creation of low conductive pathways in electron transports. Therefore, low signals of gas response for the *f*-MWCNT-PEDOT:PSS gas sensor to NH_3_ under heavy substrate-bending are represented.

The gas-sensing mechanism of the screen-printed *f*-MWCNT-PEDOT:PSS gas sensor has been proposed based on a classification of two possible mechanisms. The *f*-MWCNT-PEDOT:PSS gas sensor with a high *f*-MWCNT content has been proposed as a first possible mechanism based on a reducing reaction between chemisorbed oxygen groups on the *f*-MWCNT-PEDOT:PSS surfaces and gas molecules. The oxygen groups can be trapped on the surface of active materials in dry air at room temperature. In addition, the oxygen groups also tend to increase after functionalization with 3:1 H_2_SO_4_/HNO_3_ treatment. After the exposure of the gas-sensing film to NH_3_ vapor, the NH_3_ molecules can be adsorbed on the *f*-MWCNT-PEDOT:PSS surfaces. The reducing reaction between NH_3_ molecules and oxygen groups returns electrons to the *f*-MWCNT-PEDOT:PSS surfaces as a p-type semiconductor material. When the p-type semiconducting *f*-MWCNT-PEDOT:PSS gas sensor received electrons from NH_3_ molecules, the concentration of the hole in p-type semiconducting *f*-MWCNT-PEDOT:PSS gas sensor decreased. This leads to an increment in the electrical resistance of the *f*-MWCNT-PEDOT:PSS gas sensor after exposure to NH_3_ gas. Because of the strong bonding between the NH_3_ molecules and oxygen-containing groups, the NH_3_ chemisorbed molecules cannot be removed completely from the surface of *f*-MWCNT-PEDOT:PSS at room temperature, although the NH_3_ gas sensor is purged by the dry-air. This may lead to a creation of drift at a baseline of resistance for the *f*-MWCNT-PEDOT:PSS gas sensor with a high *f*-MWCNT content (S4 gas sensor).

With regard to the second possible mechanism, the *f*-MWCNT-PEDOT:PSS gas sensor with a low *f*-MWCNT content has been also discussed based on a direct charge-transfer process between NH_3_ molecules and *f*-MWCNT-PEDOT:PSS surfaces. Physisorption has been considered as a dominant process in the explanation of this possible mechanism. The increments in the specific adsorption area and π-π interactions can be improved by the addition of *f*-MWCNTs to PEDOT:PSS. The holes of *f*-MWCNT-PEDOT:PSS respond to donating electrons from NH_3_ molecules when they are adsorbed onto *f*-MWCNT-PEDOT:PSS surfaces. When decreasing the hole concentration, the resistance of the p-type semiconducting *f*-MWCNT-PEDOT:PSS gas sensor increases. Because physisorption is a weak π-π interaction, the gas molecules can be easily purged under dry air at room temperature. This may lead to the complete recovery of the baseline for the S0, S1, S2 and S3 gas sensors under the purging of dry-air at room temperature.

## 4. Conclusions

The *f*-MWCNTs were successfully prepared in the gas-sensing solutions by continuous stirring in PEDOT:PSS, DMSO, EG and Triton X-100. The solutions were screen-printed onto PET substrates using the low-cost system for preparation of room-temperature gas sensors and characterized for NH_3_ sensing. The optimum *f*-MWCNT content for the fabrication of an effective NH_3_ gas sensor is a 3.0% *v*/*v* solution. It presents good performance of the recovery property in dry-air flows without external excitation. The sensors were also tested under substrate-bending in different states. It can be concluded that the gas responses of the sensor under flat and weak bending did not have an obvious difference because the *f*-MWCNTs act as additional pathways in electron transport between the crack gap on the sensing films. For heavy substrate bending, the gas response of the sensor is significantly reduced due to the tensile strain effect. The gas-sensing mechanism of the screen-printed *f*-MWCNT-PEDOT:PSS gas sensor has been proposed based on a classification of two possible mechanisms such as the reducing reaction and direct charge-transfer process. This finding will be useful for the development of future electronic technology in flexibility.

## Figures and Tables

**Figure 1 micromachines-13-00462-f001:**
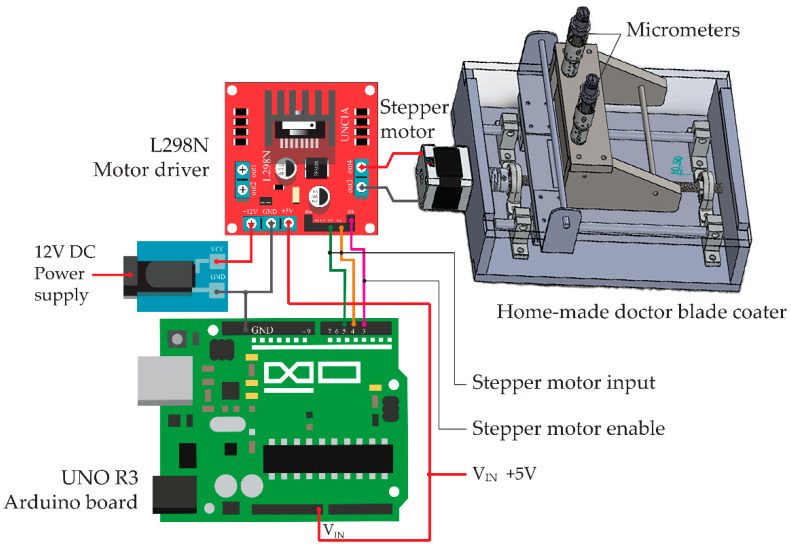
Schematic illustration of screen-printing system.

**Figure 2 micromachines-13-00462-f002:**
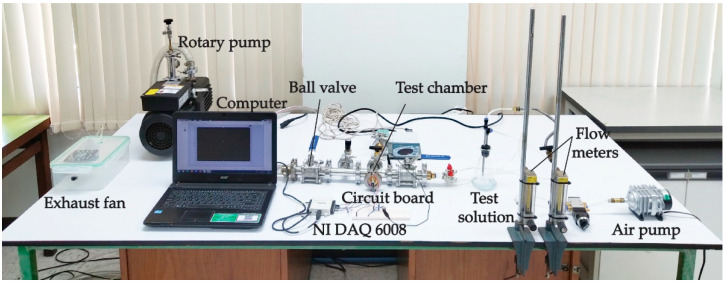
Photograph of gas-measurement system in this study.

**Figure 3 micromachines-13-00462-f003:**
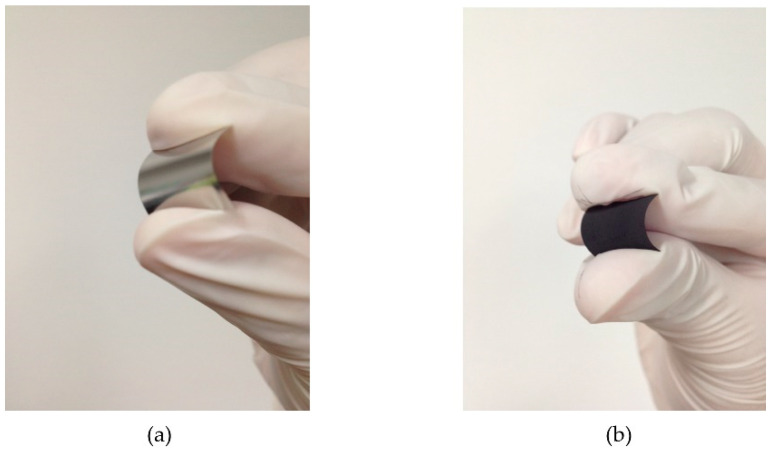
Photograph of 1.6 × 3.0 cm^2^ 304 SS foil (**a**) before and (**b**) after CVD process.

**Figure 4 micromachines-13-00462-f004:**
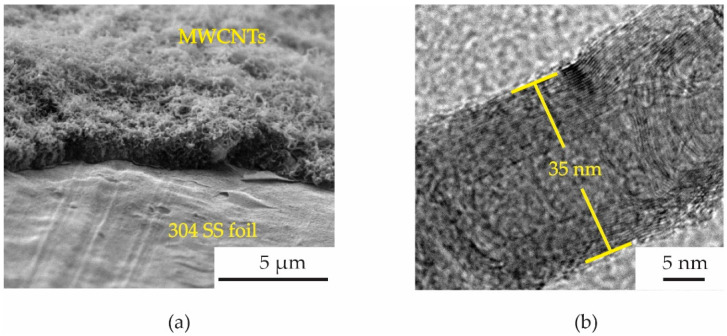
(**a**) SEM image of MWCNTs grown on 304 SS foil and (**b**) HRTEM image of synthesized MWCNT in this synthesis.

**Figure 5 micromachines-13-00462-f005:**
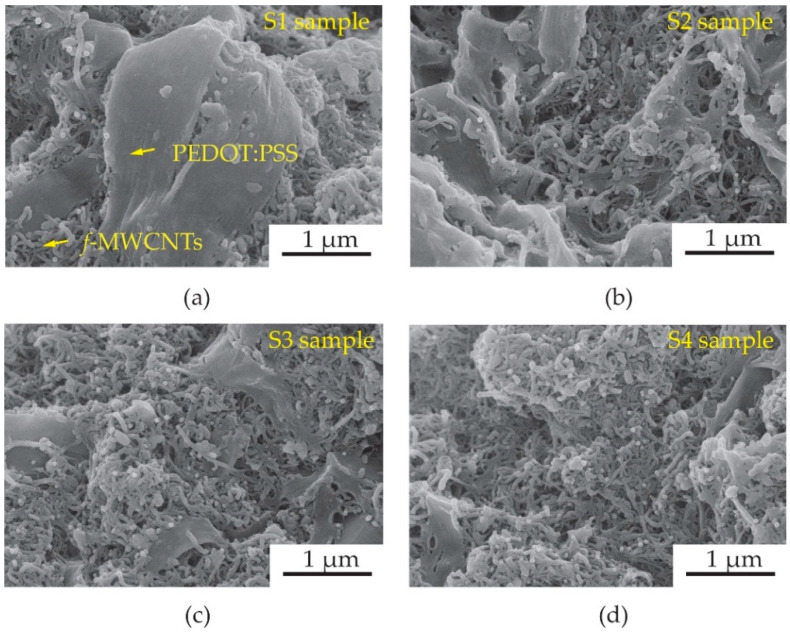
SEM images of *f*-MWCNTs embedded in PEDOT:PSS sensing-films at different samples of (**a**) S1, (**b**) S2, (**c**) S3 and (**d**) S4.

**Figure 6 micromachines-13-00462-f006:**
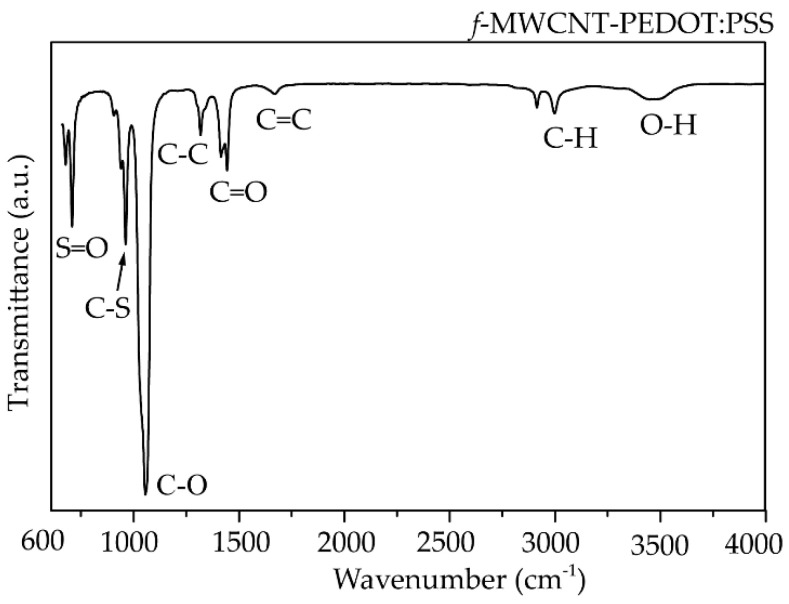
FTIR spectrum of *f*-MWCNT-PEDOT:PSS.

**Figure 7 micromachines-13-00462-f007:**
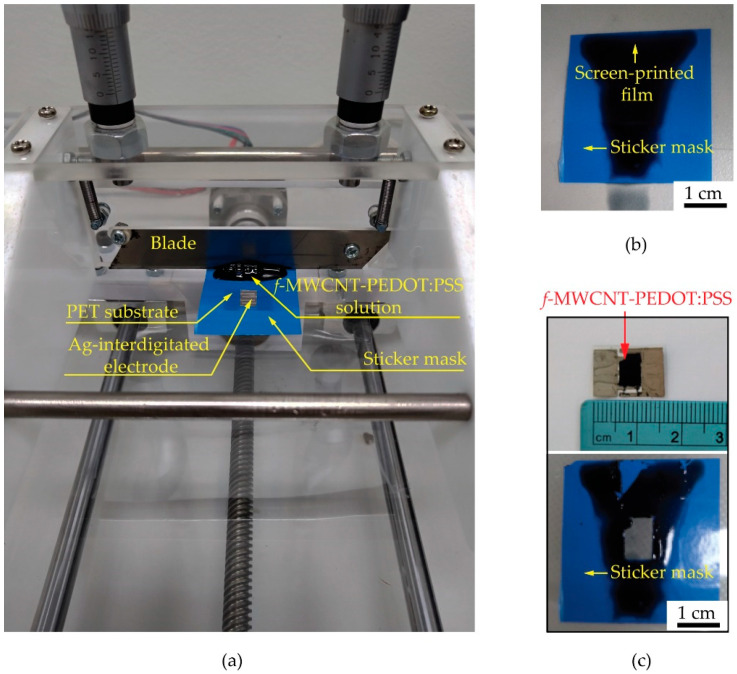
Photograph of (**a**) home-made doctor blade coater and its part. Screen-printed film of *f*-MWCNT-PEDOT:PSS gas sensor (**b**) before and (**c**) after peeling the sticker mask.

**Figure 8 micromachines-13-00462-f008:**
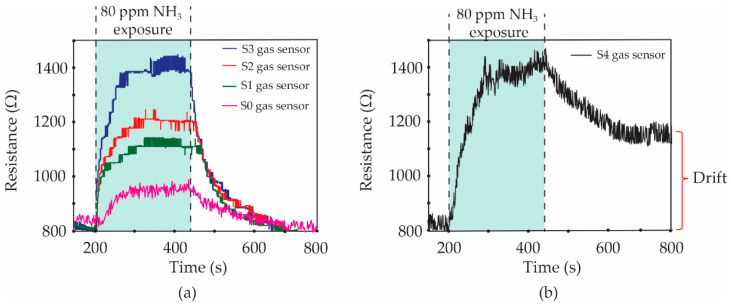
Resistance changes of (**a**) S0, S1, S2, S3 and (**b**) S4 gas sensors under 80 ppm NH_3_ exposure.

**Figure 9 micromachines-13-00462-f009:**
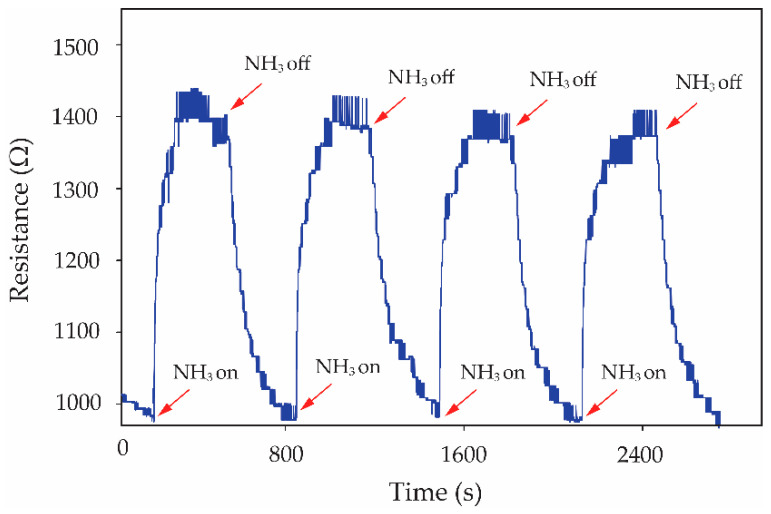
Resistance change of S3 gas sensor in handling repeated inlets of 80 ppm NH_3_ exposure.

**Figure 10 micromachines-13-00462-f010:**
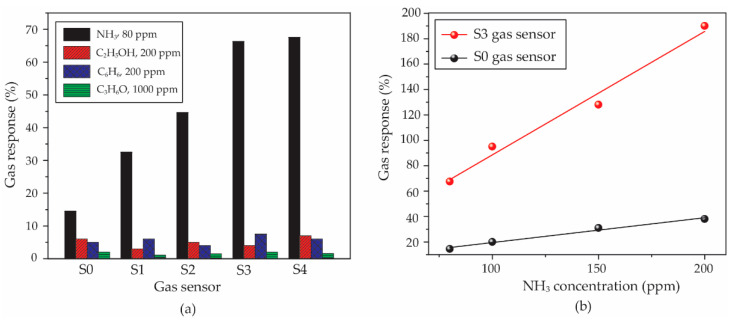
(**a**) Comparison of gas response for all sensors under different test gases. (**b**) Gas response of S0 and S3 gas sensors as a function of NH_3_ concentration.

**Figure 11 micromachines-13-00462-f011:**
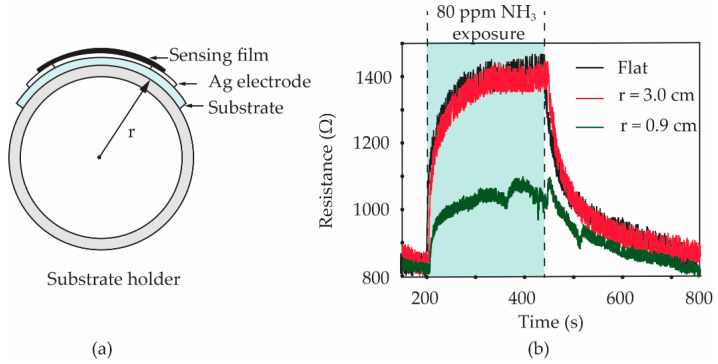
(**a**) Schematic diagram of *f*-MWCNT-PEDOT:PSS gas sensor under bending test. (**b**) Resistance changes of S3 gas sensor measured during flat and bending (r = 0.9 cm, r = 3.0 cm) states.

**Figure 12 micromachines-13-00462-f012:**
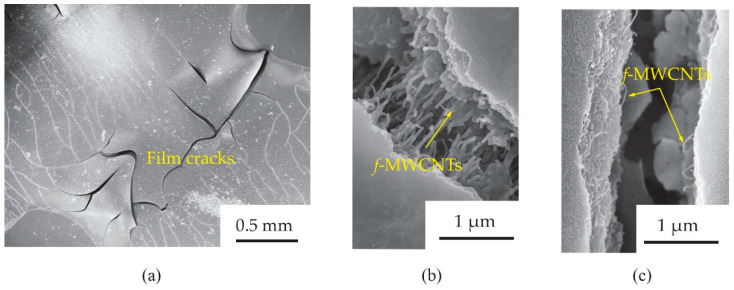
SEM images of (**a**) screen-printed *f*-MWCNT-PEDOT:PSS films (S3 sample) after bending tests with (**b**) 3.0 cm and (**c**) 0.9 cm radii.

**Figure 13 micromachines-13-00462-f013:**
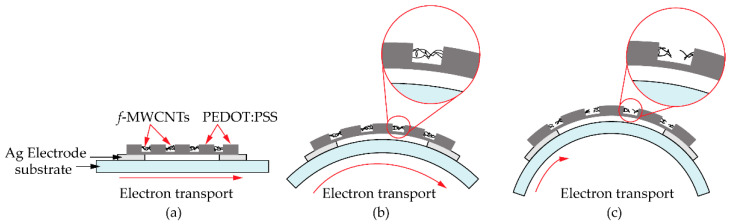
Schematic diagram of pathways in electron transports for *f*-MWCNT-PEDOT:PSS gas sensor (S3 sample) under (**a**) flat state, (**b**) 3.0 cm and (**c**) 0.9 cm bending-radius state.

**Table 1 micromachines-13-00462-t001:** Concentration of each chemicals for preparing the solutions.

Sample	*f*-MWCNT Solution (% *v*/*v*)	PEDOT:PSS (% *v*/*v*)	DMSO (% *v*/*v*)	EG (% *v*/*v*)	Triton X-100 (% *v*/*v*)
S0	0.0	90.8			
S1	0.5	90.3			
S2	2.0	88.8	5.4	3.6	0.2
S3	3.0	87.8			
S4	5.0	85.8			

## Data Availability

The data used to support the findings of this study are available from the corresponding author upon request.

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
