# Peer review of "Screen-Printing of Functionalized MWCNT-PEDOT:PSS Based Solutions on Bendable Substrate for Ammonia Gas Sensing"

_micromachines, 2022, doi:10.3390/mi13030462_

Round 1

Reviewer 1 Report

The present manuscript is dedicated to investigation of f-MWCNT-PEDOT-PSS as a sensing layer for the enhancement of gas-sensing properties. A homemade-doctor blade coater, an UNO R3 Arduino board and a L298N motor driver have been presented as a novel system for screen-printing the solutions onto the gas sensing substrates. The optimum f-MWCNT content for the fabrication of effective NH3 gas sensor was reached for 3.0% v/v solution. This finding will could be useful for developing of future electronic technology in flexibility.

All figures and captures are well presented. The mentioned literature is up to date.

So I am kindly recommending the article for publication.

Reviewer 2 Report

This manuscript presents a work to use screen-printing of functionalized MWCNT-PEDOT:PSS solutions on stainless steel substrate for Ammonia gas sensing. Their results show effectiveness of 3.0% v/v of f-MWCNT solution in detecting different concentration of NH3 gas. The work is interesting but similar concept of work has been reported back in 2014 (“MWCNT-conducting polymer composite based ammonia gas sensors: A new approach for complete recovery process”, Sensors and Actuators B: Chemical Volume 194, April 2014, Pages 213-219).

The authors would need justify what are their new insights and contribution to this field to justify the publication of this work. In addition, I would also suggest the following comments for the authors to consider:

  1. What is the effectiveness of the sensor in handling repeated inlets of NH3 gas (Fig. 8)? Would the same senor work for multiple times or is it effective only for one time use?
  2. It is good to elaborate the details about the characterization techniques, such as brand of SEM, TEM, FTIR, accelerating voltage for TEM,etc.
  3. I disagree with the schematic of f-MWCNT-PEDOT:PSS shown in Fig. 6(a), which shows that PEDOT:PSS is covering each f-MWCNT along its outer diameter. As you can see from Fig. 5(a & b), a more realistic schematic of f-MWCNT-PEDOT:PSS would be that f-MWCNT simply randomly mix up with PEDOT:PSS.
  4. It is good to have a ruler as a reference in Fig. 7(b & c) so that readers can correlate the dimension of the sensor.
  5. What does the “?” mark refer to in equation (1)?

Round 2

Reviewer 2 Report

I thank you authors for making the revision. The manuscript in much a good shape than previous version. 

The earlier publication (“MWCNT‐conducting polymer composite based
ammonia gas sensors: A new approach for complete recovery process”, Sensors and Actuators B: Chemical Volume 194, April 2014, Pages 213‐219) is quite relevant to the current work. I would suggest that authors add it as a reference and incorporate the difference between the 2 pieces of work into the manuscript.
